# Characterization of Microbial Communities from the Alimentary Canal of *Typhaea stercorea* (L.) (Coleoptera: Mycetophagidae)

**DOI:** 10.3390/insects13080685

**Published:** 2022-07-29

**Authors:** Julius Eason, Linda Mason

**Affiliations:** Department of Entomology, Purdue University, 901 West State Street, West Lafayette, IN 47907, USA; eason2@purdue.edu

**Keywords:** *Typhaea stercorea*, fungivore, alimentary canal, 16S rRNA amplicon sequencing, bacterial symbionts

## Abstract

**Simple Summary:**

Hairy fungus beetle, *Typhaea stercorea,* is a secondary post-harvest pest of stored grains that thrives by feeding on mytoxigenic fungi. Bacterial communities residing in the alimentary canal of most insects contribute to their host’s development. While there are many examples, little is known about the role of bacterial communities in the alimentary canal of *T. stercorea*. The objectives of this study were to (1) characterize the microbial communities residing in *T. stercorea* and (2) compare the microbial compositions of field-collected and laboratory-reared populations. In this study, we were able to identify bacterial communities that possess mycolytic properties and track mark changes in the microbiota profiles associated with development. The genus *Pseudomonas* was enriched in *T. stercorea* larvae compared to adults. Furthermore, field-collected *T. sterocrea* adults had a lower species richness than both larva and adult laboratory-reared *T. sterocrea*. Moreover, the gut microbial compositions of field-collected and laboratory-reared populations were vastly different. Overall, our results suggest that the environment and physiology can shift the microbial composition in the alimentary canal of *T. stercorea*.

**Abstract:**

The gut microbiomes of symbiotic insects typically mediate essential functions lacking in their hosts. Here, we describe the composition of microbes residing in the alimentary canal of the hairy fungus beetle, *Typhaea stercorea* (L.), at various life stages. This beetle is a post-harvest pest of stored grains that feeds on fungi and serves as a vector of mycotoxigenic fungi. It has been reported that the bacterial communities found in most insects’ alimentary canals contribute to nutrition, immune defenses, and protection from pathogens. Hence, bacterial symbionts may play a key role in the digestive system of *T. stercorea*. Using 16S rRNA amplicon sequencing, we examined the microbiota of *T. stercorea*. We found no difference in bacterial species richness between larvae and adults, but there were compositional differences across life stages (PERMANOVA:pseudo-F_(8,2)_ = 8.22; *p* = 0.026). The three most abundant bacteria found in the alimentary canal of the larvae and adults included *Pseudomonas* (47.67% and 0.21%, respectively), an unspecified genus of the Enterobacteriaceae family (46.60 % and 90.97%, respectively), and *Enterobacter* (3.89% and 5.75%, respectively). Furthermore, *Pseudomonas* spp. are the predominant bacteria in the larval stage. Our data indicated that field-collected *T. stercorea* tended to have lower species richness than laboratory-reared beetles (Shannon: H = 5.72; *p* = 0.057). Furthermore, the microbial communities of laboratory-reared insects resembled one another, whereas field-collected adults exhibited variability (PERMANOVA:pseudo-F_(10,3)_ = 4.41; *p* = 0.006). We provide evidence that the environment and physiology can shift the microbial composition in the alimentary canal of *T. stercorea*.

## 1. Introduction

Mycotoxins produced by pathogenic fungi have an economic impact on stored grains and pose a serious threat to food security. The mycotoxins of most concern are produced by certain species of *Aspergillus* and *Penicillium* that are commonly associated with stored grain products during processing and storage [1]. These fungal infestations are influenced by storage, environmental, and ecological conditions [2], which can affect the fungal growth on or in stored grains. Fungal contamination accounts for a large percentage (≤25%) of direct losses in grain production [3,4] and can trigger secondary insect infestations [5]. The most prevalent fungal feeding insects associated with stored grain are *Ahasverus advena* (Waltl) (Coleoptera: Silvanidae), *Cryptolestes ferrugineus* (Stephens) (Coleoptera: Laemophloeidae)*,* and *Typhaea stercorea* (L.) (Coleoptera: Mycetophagidae) [6], of which *T. stercorea* is the most common species found throughout fungal infested storage structures [7].

*Typhaea stercorea,* also known as the hairy fungus beetle, feeds on an array of fungi growing on stored grains and vectors of mycotoxigenic fungi throughout storage structures [8]. The fungi eaten by fungal feeders provide nutrients, such as sterols, that are essential for general maintenance and stimulating growth and reproduction, whereas insects lack the ability to synthesize sterols de novo [9,10]. These beetles complete their lifecycles on three fungal genera (*Aspergillus*, *Eurotium*, and *Penicillium*) that colonize stored grains [6]. Diet impacts the physiology and fitness of *T. stercorea* [6]; for example, beetles had a shorter larval development period and females laid more viable eggs when fed on *Aspergillus* followed by *Eurotium* and *Penicillum* fungal strains [6]. In addition, these beetles are able to complete their lifecycles when reared on high levels of mycotoxins produced by *A. flavus*, which are toxic to humans and other animals [6].

The insect body serves as an inclusive reservoir for microbial communities including bacteria, archaea, and fungi [11]. Diverse groups of microbial organisms can be found in most insects’ alimentary canals [12,13]. As a result, microbiota account for up to 10% of the insect’s biomass [11]. Insects and their associated microorganisms perform important functions that contribute to the degradation of organic matter, including fungi [14,15,16]. For example, two endosymbiotic bacteria, *Bacillus* and *Serratia*, secrete chitinolytic enzymes that degrade the chitin present in the cell walls of fungi, which supply nitrogen and carbon as a source of nutrients to their insect host [16,17]. In addition, several species of *Bacillus* have the capacity to express cyclic lipopeptides that inhibit the growth of certain species of *Aspergillus* [18,19]. These microbes affect the fitness of their insect hosts in different ways, such as nutrition uptake, immune defenses, and protection from pathogens [20]. Therefore, gut microbial communities tend to mediate functions that are essential to the fitness of their insect host.

In the past, it was difficult to identify microbiota and their roles within the gut microbiome. However, the advent of Next-Gen Sequencing (NGS) has allowed researchers to investigate the microbiome of insects by using amplicon sequencing. This technique is used to produce a taxonomic profile, allowing for the identification of underrepresented communities of microbes within an environment. Furthermore, this approach has been used to identify correlations between microbial communities and insect fitness [21]. The first step in understanding the interaction between gut microbes and their host is to characterize the microbiota. Measuring the diversity of the gut microbiome will serve as a starting point for understanding biological mechanisms.

The broad goal of this study was to understand the components and roles of gut microbial communities that are present in the alimentary canal of *T. stercorea*. The objectives of this study were to (1) characterize the microbial community residing in the alimentary canal of *T. stercorea* and (2) compare the microbial compositions of field-collected and laboratory-reared populations. Our hypotheses were (1) there will be a decreased microbiota composition from larval to adult life stages and (2) the gut microbiota of laboratory-reared adults and larvae will be less diverse when compared to field-collected adults. Using 16s rRNA amplicon sequencing, we examined the microbiota across all life stages, and compared diversity patterns in microbial communities of *T. stercorea* under laboratory and field conditions.

## 2. Materials and Methods

### 2.1. Typhaea stercorea Laboratory and Field Strains

*Typhaea stercorea* laboratory colonies were obtained from The Ohio State University where insect colonies were collected from spilled, moldy grain along the rail spur of a food manufacturer west of Columbus, OH, during the summer of 1986. Stock colonies were maintained on an oat/yeast/agar diet (50 g of rolled oats, 5 g of brewers’ yeast, 2 g of agar, and 15 mL of water) in glass jars (800 mL) sealed with a double layer of filter paper as a lid. Insect cultures were incubated at 30 ± 0.5 °C and 72% r.h.

Field strains of *T. stercorea* were collected from Throckmorton Purdue Agricultural Center (TPAC) using pitfall traps during the months of June and July 2018 (Appendix A). The piftall traps included a Mason jar, polyvinyl chloride (PVC) sheet, and wire rods. The Mason jars contained maize, which was infested with *A. flavus* to attract *T. stercorea*. The presence of *T. stercorea* was monitored weekly and adult *T. stercorea* were removed from the pitfall traps using an aspirator.

### 2.2. Insect Preparation and Dissection

Newly emerged late instar larvae (Figure 1A) and adults (Figure 1B) of *T. stercorea* obtained from the lab colony were collected and placed in a soufflé cup (5.5 oz) containing a cotton ball saturated with 200 µL of water, then re-treated daily with 100 µL of water for three days. After the three-day holding period, insects were randomly selected for dissection. Field-collected insects were dissected within three hours of trapping. Insect specimens were prepared for gut dissection by surface sterilization in 70% ethanol. Late instar larva and adult stages were dissected in ethanol, and the whole alimentary canal was separated from the body with the help of fine-tip forceps under microscope. Fifteen whole alimentary canals were pooled in sterile 1.5 mL centrifuge tubes containing 800 μL of ethanol and stored at −20 °C until use. A total of 5 tubes were prepared for each treatment group: laboratory-reared newly emerged late instar larva, laboratory-reared newly emerged adult and field-collected adult *T. stercorea*.

### 2.3. DNA Extraction and Sequencing of Bacterial 16S rRNA Gene

Five replicates of each group (late instar larva laboratory, adult laboratory, and adult field) contained 15 dissected tissue samples, which were used for DNA extraction (Qiagen DNeasy Blood and Tissue Kit: Valencia, CA, USA). Prior to DNA extraction, samples were centrifuged for 20 min at 15000 rpm to separate the tissues from the 70% ethanol, and then the ethanol was removed. Afterwards, the tissue was pulverized using 0.5 mL pellet pestle for 1 min. All subsequent steps were completed using the standard manufacturer’s protocol, including 4 h proteinase K digestion. Elutions were carried out with 75 μL buffer AE. The total DNA concentrations of all samples were determined on a NanoDrop 2000c Spectrophotometer (Thermo Scientific: Wilmington, DE, USA).

Samples were sent to the University of Minnesota Genomic Center (UMGC) for sequencing by high-throughput, paired-end (2 × 300 bp) sequencing, MiSeq technology (Illumina). The UMGC measured the amount of bacterial DNA present in the extracts with quantitative PCRs (qPCRs) of the bacterial 16S rRNA gene for quality control. Samples that were above 500 copy number (molecules/µL) were processed for library construction and sequencing. We used the V3F_Nextera_375F (5′-TCGTCGGCAGCGTCAGATGTGTATAAGAGACAGCCTACGGGAGGCAGCAG) and Meta_V4_806R (5′-GTCTCGTGGGCTCGGAGATGTGTATAAGAGACAGGGACTACHVGGGTWTCTAAT) universal bacterial primers to target the V3V4 region of the 16S rRNA gene of all bacteria and archaea present.

### 2.4. Sequence Curation

Demultiplexed raw sequences were extracted from the Illumina MiSeq system in FASTQ format. After removing low-quality sequences, paired-end reads were merged and curated using the Qiime2 software package v2 [22]. We imported the raw reads into Qiime2 using the format PairedEndFasqManifestPhred33, which allowed us to determine how many sequences per sample. For quality control, we used the DADA2 method to filter sequences, denoise, merge, and remove chimeras [23]. Quality filtering allowed us to trim off bases of each sequence, which removed low-quality regions of the sequences. Based on the demultiplex summary stat, we kept all 300 bp of the forward reads, but we removed 30 bp from the tail end of the reverse reads. After identifying the unique sequences and their frequency in each sample, sequences were aligned to the rRNA database project Silva_v132 and split into groups corresponding to their taxonomy at the level of genus and then assigned to operational taxonomic units (OTUs) at a 1% dissimilarity level. The analysis of composition of microbes (ANCOM) was applied to identify features that are differentially abundant across sample groups assuming that less than 25% of the features are changing between treatment groups [24].

### 2.5. Statistical Analyses

#### 2.5.1. Statistical Analyses of *T. stercorea* Life Stage Microbiota

For *T. stercorea*, life stage analysis was carried out in Qiime2. The sequence data were rarefied to a sequencing depth of 7848 sequence count (larva—lab strain (N = 4) and adult—lab strain (N = 4)). The results were plotted on an alpha–rarefaction curve using a max depth of 24,000 (Appendix A) and Permanova test was performed to distinguish the differences between life stages. Permanova is a robust non-parametric test of the general multivariate hypothesis of differences in the composition and/or relative abundances of organisms of different species in samples from different groups or treatments [25]. To characterize microbial alpha-diversity (species richness, choa 1 index, evenness, and Shannon’s diversity index), between life stages were statistically tested by Kruskal–Wallis test using the observed OTU table generated in Qiime2. The Kruskal–Wallis test is a non-parametric test used to observe the mean differences between treatments [26]. Observed OTU was used for a qualitative measure of community richness, and evenness (Pielou’s Evenness) was used to measure the community evenness [27]. Beta diversity was analyzed using the Jaccard, Bray–Curtis, unweighted UniFrac and weighted UniFrac distances to compare the different groups and plotted in a principle coordinate analysis (PCoA). The Jaccard distance is a qualitative measure of the community dissimilarity, whereas the unweighted UniFrac is also qualitative measure that incorporates phylogenetic relationships between the features (OTUs) [27]. The Bray–Curtis distance is a quantitative measure of community dissimilarity, whereas weighted UniFrac is also a quantitative measure of community dissimilarity that incorporates phylogenetic relationships between the features [27]. A *p*-value of ≤ 0.05 was used to indicate significant differences between groups.

#### 2.5.2. Statistical Analysis Comparing Field-Collected Adults to Laboratory-Reared Larvae and Adults

To evaluate if the microbiota vary between laboratory-reared and field-collected *T. stercorea*, we compared larva (N = 4) and adult (N = 4) that were reared on artificial diet in laboratory conditions to field-collected adults (N = 2). Sequences were rarefied to a sequencing depth of 66 sequences counts. The results were plotted on an alpha–rarefaction curve to max depth of 200 (Appendix A). The statistical analysis followed the procedures described above for *T. sterocera* life stage microbiota.

## 3. Results

### 3.1. Analysis of Bacterial 16S rRNA Gene Sequences

The V3V4 region of the 16S rRNA was amplified and sequenced from the alimentary canal of the larvae and adult *T. stercorea*. A MiSeq (Illumina: Minneapolis, MN, USA) instrument was used to obtain a mean of 20,773 sequences per sample. The sequences were processed and filtered through the QIIME2 pipeline [22], and a total of 112 unique operational taxonomic units (OTUs) were obtained among the samples. 

### 3.2. Alpha and Beta Diversity of T. stercorea

Observed OTUs and Chao1 were used to measure richness within a sample, while evenness measured the relative abundance of species richness (Figure 2). The Shannon Index was used to measure alpha diversity by measuring the number of different species and their relative abundance within a sample. The analyses of *T. stercorea* microbiota diversity showed that species richness using both observed OTUs and Chao 1 was not significantly different across life stages (Figure 2A,C). The adults showed larger variation in species richness, which had higher and more constant value when compared to larvae. In addition, evenness exhibited no significant difference between life stages (Figure 2B). The alpha diversity using the Shannon Index revealed that there were no significant differences across life stages (Figure 2D). Although larva samples showed more variation, adult samples had higher and more constant values. The beta diversities (Jaccard, Bray–Curtis, unweighted UniFrac and weighted UniFrac) were used to measure the gut microbiota composition. These dissimilarity matrices were observed in a principle coordinates analysis plot, where samples were separated according to their life stage (Figure 3A,D). PERMANOVA was calculated using distance matrices (Jaccard and Bray–Curtis), which indicated that the gut microbiota composition was significantly different between life stages (Table 1, *p < 0.05*). However, unweighted UniFrac and weighted UniFrac analyses exhibited no significant differences in species composition between larva and adult *T. stercorea*, suggesting that the phylogenetic distances between observed OTUs were weakening the beta diversity (Table 1, *p > 0.05*).

### 3.3. Microbial Taxonomic Composition in the Alimentary Canal Shows Differences between Life Stages of Typhaea stercorea

According to the Silva v.132 database, there were 112 OTUs aggregated into 38 genera and 8 phyla. Of these phylotypes, 97.81% belong to the phylum Proteobacteria (data not shown). The two most abundant genera in the larvae and adults were an unspecified genus of the Enterobacteriaceae family (46.60% and 90.97%, respectively) and *Enterobacter* (3.89% and 5.75%, respectively) (Figure 4A). The analysis of composition of microbiomes (ANCOM) was conducted to identify the taxa that was differentially abundant across life stages of *T. stercorea*. With the use of ANCOM, we observed that the genus *Pseudomonas* was significantly different in the gut composition between larvae and adults (47.67% and 0.21%, respectively) (Figure 4A). Larvae had the most genera present at greater than 1%, with three genera identified, and adults had two genera (Figure 4A).

In order to describe which OTUs were present in both larva and adult life stages, an OTU was assumed to be present if it was observed in at least two of replications in each life stage. The data show that adults have higher numbers of OTUs present in their alimentary canal compared to larvae (Figure 4B). The intersection between the larva and adult stages of *T. stercorea* shares four OTUs throughout their lifecycle (Figure 4B). The shared OTUs belong to the genera *Enterobacter*, *Streptomyces*, *Staphylococcus,* and an unspecified genus belonging to the family Enterobacteriaceae.

### 3.4. Variation in Diversity and Microbial Composition between Laboratory-Reared and Field-Collected Populations of T. stercorea

The species richness of laboratory-reared (larva and adult) and field-collected populations of *T. stercorea* was not significantly different (Chao 1 Index: Kruskal–Wallis *p* = 0.11) (Figure 5A)*,* although the laboratory-reared insects exhibited a trend of higher species richness than field-collected specimens. Field-collected adults tended to have a lower alpha diversity using the Shannon Index compared to laboratory-reared larvae and adults (Kruskal–Wallis (all groups): H = 5.72; *p* = 0.057) (Figure 5B). The species diversity of laboratory-reared larvae was not significantly different when compared to laboratory-reared adults (Kruskal–Wallis (pairwise: H = 2.08; *p* = 0.148), which it corroborated the previous objective. The Jaccard, Bray–Curtis, and weighted UniFrac analyses showed three distinct clusters, suggesting that the microbial compositions in the alimentary canal between laboratory-reared and field-collected populations are different (Figure 6 and Table 2). The unspecified genus of the Enterobacteriaceae family was still the dominant taxa, representing 27.4% and 90.5% of the microbiota in laboratory-reared larvae and adults, respectively, and 50.0% in field-collected adults (Figure 7). The field-collected adults had the most genera present at greater than 5%, with seven genera identified (Figure 7). Of these seven genera, *Apibacter*, *Alcaligenes*, and *Enterobacter* were identified at the genus level, while the other phylotypes were classified as unspecified genera in the Bacillaceae, Bacillales, Enterobacteriaceae, and Weeksellaceae families.

## 4. Discussion

Numerous studies have focused on identifying and characterizing microbial communities from the alimentary canal of insects that include, but are not limited to, aphids, bees, cockroaches, termites, and thrips [28]. While many microbial communities in the alimentary canals of insects have been described, many still await characterization, especially with respect to stored product insects. Therefore, we describe and have analyzed a new, important aspect of this species of economically important stored product insects. Previous work has identified microorganisms that exhibit a wide diversity of specialized interactions with their hosts, relating to nutrients, growth, development and other physiological processes.

*Typhaea stercorea* (L.) is an important pest of stored products that serves as an indicator of increases in fungal biomass during storage, leading to grain quality loss. Many pathogenic fungi found on stored grain products can produce mycotoxins which cause a variety of adverse health threats to both humans and livestock. *Typhaea stercorea* actively feeds on these pathogenic fungi and mycotoxins with no apparent adverse effects. This suggests that the gut microbiota metabolizes fungal diets and, in return, provides sterols and other nutrients that facilitate the growth and development of *T. stercorea*. Many insect-associated microorganisms promote an insect’s capacity to utilize diets of low or unbalanced nutritional content by providing specific nutrients that the insect cannot synthesize, including amino acids, vitamins, and sterols [11].

Here, we surveyed the microbial composition in the alimentary canal of *T. stercorea* as a first step towards understanding how the gut microbiota may play a role in growth and development. The initial hypothesis was that there will be a decrease in bacterial diversity in the gut composition from the larval to adult life stages. We showed that there is low bacterial diversity found in the alimentary canal of *T. stercorea* across life stages. From the larval to adult stages, the gut bacterial community is potentially purged from the onset of the radical remodeling of the alimentary canal during metamorphosis (Figure 1C,D), which has been observed in other holometabolous insects [29]. For alpha diversity, the observed OTUs and Shannon Index showed no differences across life stage, indicating that the distribution of OTUs from *T. stercorea’s* environment remains constant. In contrast to the lack of alpha diversity, we found differences between the beta diversity of the gut microbiota and life stages of *T. stercorea*, which suggests that diet or radical remodeling of the gut morphology/physiology are two of the most important factors that influence the assemblage of the gut microbiota [13,30,31,32,33,34]. Throughout *T. stercoreae’s* development, larvae possess two unique genera, *Pseudomonas* and *Virgibacillus*, whereas adults had four rare genera *Brachybacterium*, *Brevibacterium*, *Lawsonella*, and *Oceanobacillus*. Furthermore, four phylotypes (*Streptomyces*, *Staphylococcus*, *Enterobacter* and an unidentified genus belonging to the family Enterobacteriaceae) were shared across life stages. Of these diverse genera found in the alimentary canal of *T. stercorea*, *Brevibacterium*, *Enterobacter*, *Pseudomonas*, *Staphlococcus*, and *Streptomyces* are known to exhibit antagonistic effects against pathogenic fungi (e.g., *Aspergillus* spp, *Fusarium* spp, and *Penicillum* spp) [19].

For all developmental stages of *T. stercorea*, the most dominant phylum was Proteobacteria (%), followed by Actinobacteria and Firmicutes. A microbiome study of the fungivore *Hoplothirps carpathicus* (Thysanoptera: Phlaeothripidae) found similar results, i.e., that the microbiome of *H. carpathicus* was also dominated by the phylum of Proteobacteria (57.49%). Kaczmarcyzk et al. [35] also noted that there was an increase (>2-fold) in the phylum Proteobacteria during developmental stages from pupa to adult. Other studies have indicated that the phylum Proteobacteria shows enriched diversity in adults compared to larvae [35,36,37]. However, our findings did not agree with these previous studies, since the *T. stercorea* larvae and adults had similar relative abundances of the phylum Proteobacteria throughout their lifecycles.

There is now persuasive evidence that insects that predominately feed on fungal diets are host to unique bacterial communities with mycolytic properties [38,39]. Mycolytic bacterial communities belong to groups of the phyla Actinobacteria, Bacteroidetes, Firmicutes, and Proteobacteria [28,40,41,42]. These phyla produce key enzymes which exhibit chitinase activity that supports a shift to fungal diets [38,43,44]. These enzymes (e.g., α-mannanases, β-1,3/1,6-glucanases, and chitnases) target the cell walls of fungi composed of complex and dynamic structures of mannan, glucan, and chitin [42,45]. Moreover, these mycolytic enzymes, such as β-1,3-glucananase, serve addition roles by protecting their host from fungal infections [46]. Their relative abundance in the alimentary canal may be caused by the digestion of a protein-rich fungal diet [44,47]. Similarities in host diet have been shown to drive convergence in the functional potential of gut microbes in other insects [38]. The degradation of chitin in nature is primarily carried out by bacterial taxa, such as pseudomonads, enteric bacteria, gliding bacteria, actinomycetes, and members of the genera *Bacillus*, *Vibrio*, and *Clostridium* [48,49].

Our study has allowed us to track marked changes in microbiota profiles associated with development. The gut microbiota composition was different between life stages. We found that the genus *Pseudomonas* was enriched in *T. stercorea* larvae as opposed to adults. *Pseudomonas* is classified as a Gram-negative bacterium that is commonly found in a variety of environments (e.g., insects, soil, and water). Previous literature states that the presence of *Pseudomonas melophtora* is necessary for larval survival and development in various stages of the apple maggot, *Rhagoletis pomonella* (Diptera: Tephritidae) [50]. *Pseudomonas* has been shown to synthesize amino acids, such as methionine and cystine, which are required for the development and growth of their insect host [51]. Other studies have shown that *Pseudomonas savastanoi* produces methionine and threonine, which are required for the olive fly, *Dacus oleae* (Diptera: Tephritidae), to complete its lifecycle from larva to adult [52]. Therefore, we suggest that the genus *Pseudomonas* is a facultative secondary endosymbiont that resides in the alimentary canal and breaks down the chitin walls of fungi, providing nutrients to its host. In this case, *Pseudomonas* synthesizes amino acids that may be rare in the fungal diet of their host, *T. stercorea*. In addition, other strains of *Pseudomonas* can metabolize insecticides [53,54], such as neonicotinoids, which suggests that *Pseudomonas* species perform protective functions during the developmental stages of *T. stercorea*.

Lastly, we compared the microbiota from the alimentary canal of laboratory strain specimens (larvae and adults) to field-collected *T. stercorea* adults. We hypothesized that the gut microbiota of reared larvae and adults will lack diversity compared to field-collected adults. Our data indicate that field-collected *T. stercorea* tend to have lower species richness than laboratory-reared beetles. In contrast, other studies have demonstrated that the gut microbiota of field-collected insects typically possess more diverse bacterial species than laboratory-reared insects [32,55,56]. There was no clear explanation as to why our study showed that field-collected *T. stercorea* tended to have lower species richness than laboratory-reared members. However, it has been suggested that ecological exogenous factors can influence microbiota diversity, which indicates that the host’s diet and habitat may affect the insect gut microbiota’s species richness and composition [55]. As expected, the microbial gut composition of adult field populations was different to those of laboratory-reared larvae and adults. Field populations possessed several known chitinolytic bacteria belonging to the families Bacillaceae and Enterobacteriacea. Furthermore, the family Enterobacteriacea, which belongs to the phylum Proteobacteria, showed the highest relative abundance in laboratory-reared adults and field-collected adults. Our findings thus suggest that insects associated with fungal communities depend on the family Enterobacteriaceae in the adult stage. A previous study indicated that the family Enterobacteriaceae was one of the dominant bacterial families in the gut microbiota of the bark beetles *Dendroctonus* spp. Interestingly, the genus *Enterobacter* contributes to host nutrition by fixing atmospheric nitrogen [57,58,59]. Based on previous reports, several chitinolytic bacterial strains of Bacillaceae, Enterobacteriaceae, and Pseudomonadaceae families were classified as antagonistic microbes towards pathogenic fungi. Thus, it is imperative to continue to study these chitinolytic microbes and their properties for potential use as a biological control of pathogenic fungi associated with stored grain products.

In conclusion, this report presents data from a profile analysis of gut bacterial communities of *T. stercorea* through the use of NGS 16S rRNA amplicon sequence data. The current study adds to our understanding of how important mutualistic prokaryotes found within the gut microbiota may provide essential nutrients (e.g., sterols, vitamins, carbohydrates, and amino acid synthesis/metabolism) during the growth and development of *T. stercorea*. For future investigations, manipulating bacterial communities through the use of antibiotics will allow for the testing of emerging hypotheses on the role of gut microbes in their host’s lifecycle and fecundity.

## Figures and Tables

**Figure 1 insects-13-00685-f001:**
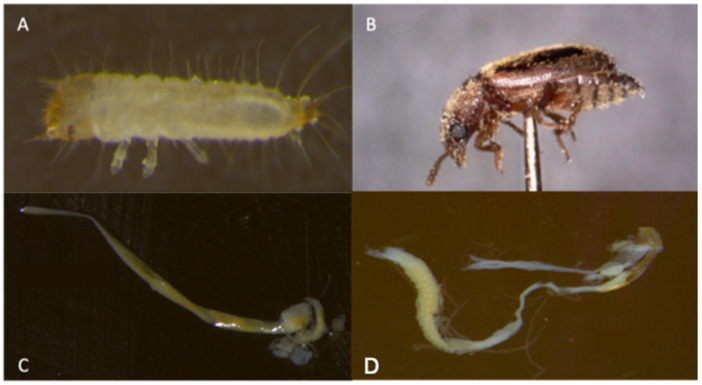
Different life stages of *T. stercorea*, including their alimentary canal. (**A**) Late instar larva. (**B**) Adult. (**C**) Alimentary canal of late instar larva. (**D**) Alimentary canal of adult.

**Figure 2 insects-13-00685-f002:**
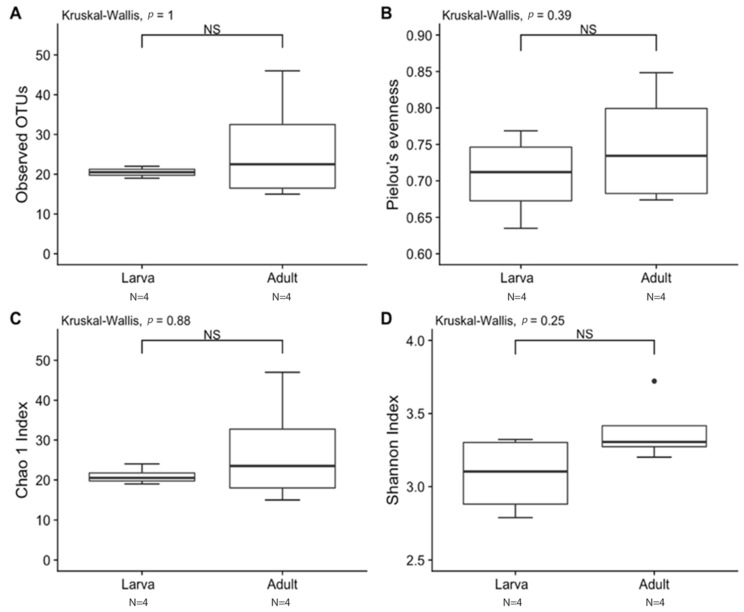
Alpha diversity between laboratory-reared larva and adult *T. stercorea* gut microbiota. A *p*-value of ≤ 0.05 was used to indicate significant differences between groups. NS denotes non-significant differences between the groups. N denotes the sample size. (**A**) Species richness boxplot. (**B**) Pielou’s evenness boxplot. (**C**) Chao 1 Index boxplot. (**D**) Shannon Index boxplot. For each group, the bars delineate the means, the hinges represent the lower and upper quartiles, the whiskers extend to the most extreme values, and black dots (●) denote outliers are plotted if present.

**Figure 3 insects-13-00685-f003:**
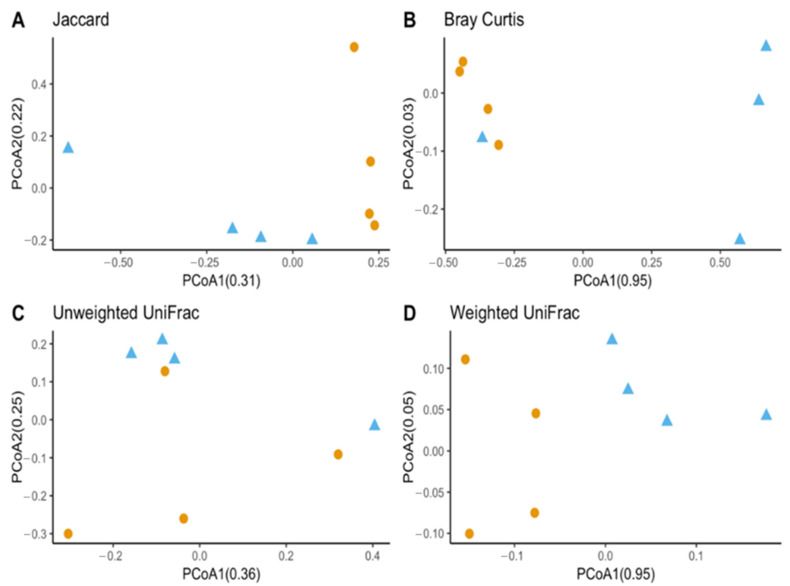
Beta diversity between laboratory-reared *T. stercorea* larva and adult gut microbiota composition. (**A**) Jaccard PCoA graph showing PCoA1 (0.31 variation) and PCoA2 (0.22 variation). (**B**) Bray–Curtis PCoA graph showing PCoA1 (0.95 variation) and PCoA2 (0.02 variation). (**C**) Unweighted UniFrac PCoA graph showing PCoA1 (0.36 variation) and PCoA2 (0.25 variation). (**D**) Weighted UniFrac PCoA graph showing PCoA1 (0.95 variation) and PCoA2 (0.05 variation). Blue triangles (▲) denote larvae and orange circles (●) denote adults.

**Figure 4 insects-13-00685-f004:**
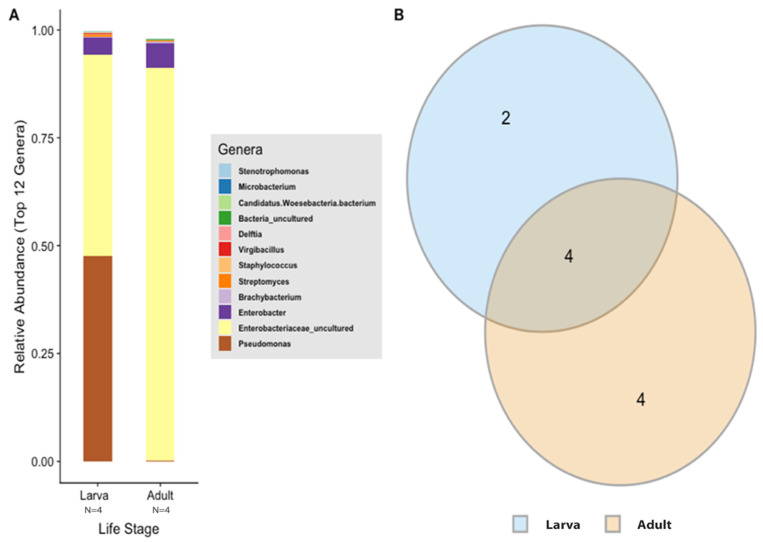
Gut microbiota composition of *T. stercorea*. (**A**) Taxonomy graph comparing the relative abundances of genera present between larva and adult *T. stercorea*. The 12 most abundant genera are shown. (**B**) Two-part Venn diagram comparison between laboratory-reared larva and adult of *T. stercorea* gut microbiota, showing the OTUs shared among life stages: larva (blue) and adult (orange). The numbers in the diagrams represent how many OTUs were unique in life stages or shared between life stages as their areas intersect.

**Figure 5 insects-13-00685-f005:**
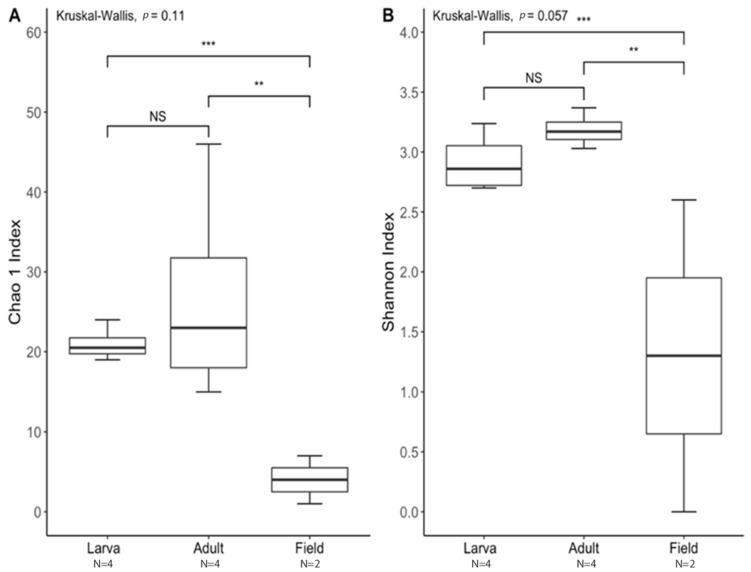
Alpha diversity between laboratory-reared larvae and adults vs. field-collected *T. stercorea* adults. A *p*-value of ≤ 0.05 was used to indicate significant differences between groups. ** denotes significant differences between laboratory-reared adult and field-collected adult. *** denotes significant differences between laboratory-reared larva and field-collected adult. NS denotes non-significant differences between groups. N denotes the sample size. (**A**) Chao 1 Index. (**B**) Shannon Index. For each group, the bars delineate the means, the hinges represent the lower and upper quartiles, the whiskers extend to the most extreme values, and outliers are plotted if present.

**Figure 6 insects-13-00685-f006:**
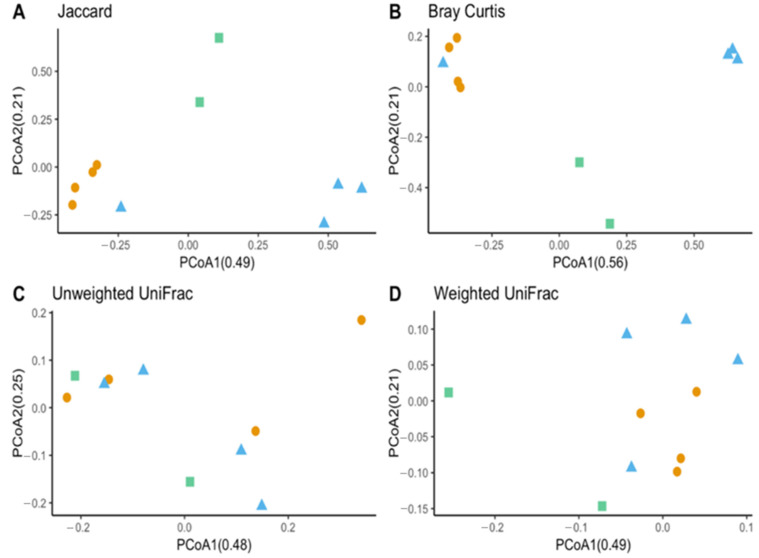
Beta diversity between laboratory-reared *T. stercorea* larva and adult gut microbiota composition vs. field-collected adult. (**A**) Jaccard PCoA graph showing PCoA1 (0.48 variation) and PCoA2 (0.25 variation). (**B**) Bray–Curtis PCoA graph showing PCoA1 (0.56 variation) and PCoA2 (0.21 variation). (**C**) Unweighted UniFrac PCoA graph showing PCoA1 (0.48 variation) and PCoA2 (0.25 variation). (**D**) Weighted UniFrac PCoA graph showing PCoA1 (0.49 variation) and PCoA2 (0.21 variation). Blue triangles (▲) denote laboratory-reared larvae, orange circles (●) denote laboratory-reared adults, and green squares (■) denote field-collected adults.

**Figure 7 insects-13-00685-f007:**
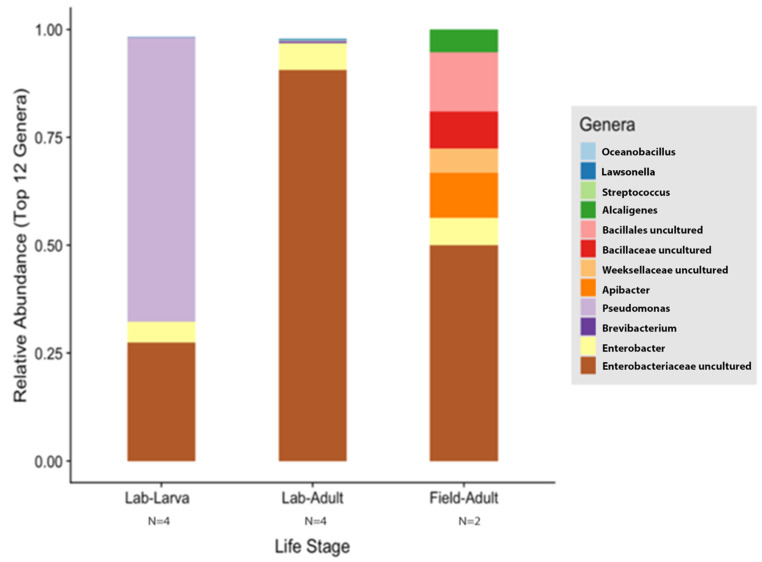
Comparison of laboratory-reared larvae and adults vs. field-collected adult gut microbiota composition of *T. stercorea*. Taxonomy graph comparing the relative abundance of genera present in *T. stercorea*. The 12 most abundance genera are shown.

**Table 1 insects-13-00685-t001:** PERMANOVA analysis based on distance matrices between laboratory-reared *T. stercorea* larvae and adults.

Beta Diversity Measure (PERMANOVA)	Pseudo-F	*p*-Value
Jaccard	1.91	0.02
Bray–Curtis	8.22	0.03
Unweighted UniFrac	1.33	0.26
Weighted UniFrac	8.05	0.06

A *p*-value of < 0.05 was used to indicate significant differences between groups.

**Table 2 insects-13-00685-t002:** PERMANOVA analysis based on distance matrices between laboratory-reared larvae and adults of *T. stercorea* gut microbiota vs. field-collected adults.

Beta Diversity Measure (PERMANOVA)	Pseudo-F	*p*-Value
Jaccard	4.23	0.002
Bray–Curtis	4.41	0.006
Unweighted UniFrac	1.05	0.41
Weighted UniFrac	3.70	0.01

A *p*-value of < 0.05 was used to indicate significant differences between groups.

## Data Availability

Data are contained within the article or Appendix A. Raw data are available upon request from the corresponding author.

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
