# Peer review of "Characterization of Microbial Communities from the Alimentary Canal of Typhaea stercorea (L.) (Coleoptera: Mycetophagidae)"

_insects, 2022, doi:10.3390/insects13080685_

Round 1

Reviewer 1 Report

 Dear authors:

Within the presented study, you have characterized the microbial community in T. stercorea alimentary canal and compared the microbial composition of field collected and laboratory reared populations. The work undertaken is relatively well planned with methods corresponding to what you were looking for; results are well reported with appropriate discussion. For the above mentioned considerations the paper is pertinent for the journal and worthwhile to be seriously considered for publication. Anyway some points should be arising up to the authors.

Comments:

1. In my opinion and experience, the threshold for statistical significance is set too high. The most common threshold is p ≤ 0.05. I am aware that the threshold depends on your field of study and I have come across thresholds of 0.01, or even 0.001, but have no experience of such high threshold. Can authors please explain why did they use such high p value. However, I strongly suggest if it is possible, to use lower threshold.

2. Line 122. Why did authors pool alimentary canals of 15 individuals instead of analyse them separately within groups like they did for statistical analysis comparing field-collected adults and laboratory-reared larvae and adults?  If it was due to low DNA yield (due to the small amount of material), maybe authors could have pooled 5 individuals, and then have 3 biological replicates per group. In line 129, when mentioning five replicates, these are then technical replicates. Please explain.

3. When motioning programs and platforms used, please provide references.

Reviewer 2 Report

This study characterized the microbial communities residing in hairy fungus beetle of different age and compared the microbial composition of field-collected and laboratory-reared adults using 16S rRNA amplicon sequencing. This research revealed some differences between those treatments.

There are my concerns and suggestions:

(1)   The authors claimed that there was no difference in bacterial species richness between larvae and adults, but there were compositional differences across life stages. I did find the results across life stages. There are only larva and adult stage, are not there?

(2)   I suggested to present the name of the top three abundant bacteria in each treatment in the Abstract.

(3)   Pseudomonas spp. was differentially abundant across life stages. Why not to say Pseudomonas spp. are the predominant bacteria in larva stage but not in adult stage?

(4)   Our data indicated that field collected T. stercorea tended to have lower species diversity than laboratory-reared beetles. Here, species diversity or community diversity? In general, field collected insects have higher bacterial diversity than laboratory-reared insects. Can the authors give some explanations?

(5)   Diet is a key factor shaping the bacterial community in the gut microbiome of insects. But the laboratory-reared insects were only fed with one type of food and the food of field collected insects are unknown. This may be the biggest deficiency of this paper.

(6)   Please pay attention to the writing format of Latin scientific names, such as Italics.

(7)   How many replicates were used? Is the number of replicates of each process consistent?

(8)   Figure2, why not to present the p values on the top of the Horizontal lines.

(9)   (A) Species richness boxplot (B) Pielou’s evenness boxplot (C) Chao 1 Index boxplot (D) Shannon Index boxplot. I think this sentence is missing commas.

(10) Figure 4. Larvae is in the plural form, but adult is in singular form. Why? There are replicates in each treatment, but why don't the replicates appear in the histogram. I strongly recommend adding replicates to the histogram.

Round 2

Reviewer 1 Report

The authors have addressed all comments and accordingly introduced changes in the manuscript. Thus,  the manuscript  is acceptable for publication in the present form.

Reviewer 2 Report

Compared with the previous version, the quality of the manuscript has been greatly improved. I suggest that this manuscript be accepted for publication. But it is necessary for authors to read through the full text to eliminate small mistakes.